# Artichoke by Products as a Source of Antioxidant and Fiber: How It Can Be Affected by Drying Temperature

**DOI:** 10.3390/foods10020459

**Published:** 2021-02-19

**Authors:** Ariel A. Borsini, Beatriz Llavata, Mónica Umaña, Juan A. Cárcel

**Affiliations:** 1Laboratorio de Preservación y Envases, Facultad de Ciencias Exactas, Químicas y Naturales, Universidad Nacional de Misiones, N3304 Posadas, Misiones, Argentina; arielborsini@gmail.com; 2Group of Analysis and Simulation of Agri-food Processes, Food Technology Departament, Universitat Politècnica de València, 46022 Valencia, Spain; beallaca@etsiamn.upv.es; 3Department of Chemistry, University of the Balearic Islands, Ctra, Valldemossa, km 7.5, 07122 Palma de Mallorca, Spain; monica.umana@uib.es

**Keywords:** phenolic content, antioxidant capacity, Vitamin C, alcohol insoluble residue, swelling capacity, fat absorption capacity, water retention

## Abstract

The reuse of food industry by-products constitutes one of the essential pillars of the change from a linear to a circular economic model. Drying is one of the most affordable techniques with which to stabilize by-products, making their subsequent processing possible. However, it can affect material properties. The objective of this study was to assess the effect of the drying temperature on the drying kinetics and final quality of the main artichoke processing by-products, bracts and stems, which have never been studied as independent materials. For this purpose, air drying experiments at different temperatures (40, 60, 80, 100 and 120 °C) were carried out. The alcohol insoluble residue (AIR) and the total phenolic content (TPC), antioxidant capacity (AC) and vitamin C (VC) of the fresh and dried samples were determined. The bracts dried faster than the stems, increasing drying rate with temperature. The two by-products presented relatively large amounts of AIR, the content being higher in bracts, but better functional properties in stems. The TPC, AC and VC values of the dried samples decreased in relation to the fresh samples, with the temperatures of 40 °C (bracts) and 120 °C (stems) being the most adequate for the purposes of preserving these characteristics.

## 1. Introduction

The Mediterranean diet is particularly rich in vegetables with active compounds, contributing to its positive effect on human health [1]. One example is artichoke (*Cynara scolymus* L.), whose beneficial effects can be mainly attributed to components with antioxidant properties, such as mono- and dicaffeoylquinic acid, caffeic acid, sesquiterpene and volatile flavonoids, as well as its high inulin and fiber content [2]. The internal soft bracts and the soft portions of the bud, commonly called artichoke hearts, are the proportions consumed fresh, canned or frozen [3]. This means that the vegetable processing industry produces large amounts of waste biomass (80–85%) mainly constituted by external bracts and stems [4]. These by-products have been shown to be a potential sources of antioxidant compounds and dietary fiber, mainly pectin and inulin [5,6]. In fact, artichoke by-product extracts have been used as fiber supplement in the functionalization of products [7,8], in the extraction of antioxidants [9] and even in the recovery of polyphenols or bioenergy production [10]. However, the use of artichoke by-products needs a prior stabilization process, which usually includes blanching and drying [11]. The aim of blanching is to avoid enzymatic degradation reactions, while drying reduces the water activity of the material, decreasing the microbiological action and limiting the chemical and physical changes during storage. Moreover, drying leads to a substantial reduction in mass and volume, minimizing packaging, storage and transportation costs [12]. The most commonly used drying technique is hot air drying [13] due to its simplicity and the ease control of process compared with other more complex or expensive techniques. In this case, the water transport from a material fundamentally depends on the external conditions of temperature, relative humidity, velocity and the direction of the hot air flow, as well as the characteristics and geometry of the solid [14]. Modeling of drying kinetics permits the objective quantification of the influence of process variables in the rate of the process and it is essential to optimize the operation. The use of theoretical models, such as those based in Fick’s law of diffusion, is preferred to the empirical ones because they can be useful to predict moisture content evolution in the different conditions that they were fitted, e.g., different sample geometry or initial moisture content.

The most-used technique to intensify hot air drying is the increase of drying temperature. However, the influence of temperature on the drying rate depends on the products and sometimes, shortening the drying time could not compensate the increase of energy consumption, which the increase of temperature causes. Moreover, the thermal treatment promotes organoleptic and nutritional changes in food products that may cause quality degradation [15] or even the formation of new compounds [16]. According to the author’s knowledge, the drying of the main artichoke by-products, bracts and stems, has never been studied independently. These by-products have been shown to have different composition and features [17,18]. However, they have always been treated as a single material [6,10,19,20]. Therefore, the objective of this study was to separately assess the influence of the drying temperature on the drying kinetics and on some quality related parameters (fiber and antioxidant properties) of both artichoke by-products, external bracts and stems. In this sense, better drying conditions could be identified, which can improve their uses as functional ingredients or fiber/antioxidant supplements.

## 2. Materials and Methods

### 2.1. Raw Material

Fresh artichokes (*Cynara scolymus* L.) were acquired in the Central Market of Valencia, Spain. The pieces selected were of uniform size and color and were stored at 4 ± 1 °C until the experiments.

### 2.2. Drying

The bracts and stems were separated from the artichokes to simulate the waste produced by the industrial processing of artichokes and independently processed. In the case of bracts, only the external ones were considered extracting a number of 20–25 bracts from each artichoke. The stems were transversally cut to obtain homogeneous cylindrical samples of 15.0 ± 0.5 mm in height (3–4 by artichoke). For each drying experiment, the samples (bracts or stems) were placed in aluminum trays and introduced into a forced convection drying chamber (Model FD 260, Binder, Tuttlingen, Germany). The air drying tests were performed at atmospheric pressure using an air velocity of 1 m/s and at five different temperatures (40, 60, 80, 100 and 120 °C). This range of temperature and air velocity is usually applied in the drying of agro-food products [13,14,15]. The relative humidity remained below 10% in every experiment carried out. At pre-set intervals (10 min), samples were taken out from the chamber and weighed. The experiments were extended until the samples reached constant weight (percentage of weight variation lower than 2% in three consecutive measurements). All the drying conditions were replicated at least in triplicate. The initial and dried sample moisture contents were measured following the standard method 934.01 (AOAC, 1997). After the drying process, the samples were milled in a grinder (Blixer 2, Robotcoupe, Vincennes, France), sieved (particle size under 200 μm) and vacuum packed until further analysis.

### 2.3. Modeling of Drying Kinetics

A diffusive model based on Fick’s diffusion law was considered in order to mathematically describe the drying kinetics. The corresponding governing equations for laminar (bracts) and cylindrical (stems) geometry, considering the effective diffusivity (*D_eff_*) as constant, were:(1)$$\frac{\partial W\left(x,t\right)}{\partial t}=D_{eff}\frac{\partial^{2}W\left(x,t\right)}{\partial x^{2}}$$
(2)$$\frac{\partial W\left(r,t\right)}{\partial t}=D_{eff}\left(\frac{\partial^{2}W\left(r,t\right)}{\partial x^{2}}+\frac{1}{r}\frac{\partial W\left(r,t\right)}{\partial r}\right)$$
where *W* is the local moisture content (kg water/kg dry matter (d.m.)); *t* is time (s); *D_eff_* is the effective diffusivity (m^2^/s); x and *r* the characteristic transport direction (m) for laminar and cylindrical geometry, respectively.

To solve Equations (1) and (2), it was considered that the initial moisture content was homogeneous throughout the samples, the solids were symmetric and their volume remained constant during drying. The external resistance to mass transport was included in the model by the boundary conditions (Equations (3) and (4)):(3)$$-D_{eff}\rho_{ss}\frac{\partial W\left(x,t\right)}{\partial x}=k\left(a_{w}\left(x,t\right)-\varphi_{air}\right)$$
(4)$$-D_{eff}\rho_{ss}\frac{\partial W\left(r,t\right)}{\partial r}=k\left(a_{w}\left(r,t\right)-\varphi_{air}\right)$$
where *a_w_* is the water activity of the samples; *ρ_ss_* is the density of the dry solid (kg d.m./m^3^); *φ_air_* is the relative humidity of the drying air; *k* is the mass transfer coefficient (kg water/m^2^s). An implicit finite difference method was chosen to estimate the parameters of the model, *D_eff_* and *k*, using Matlab 2015 (The Mathworks, Inc., Natick, MA, USA).

The goodness of the fit was evaluated from the percentage of the explained variance (%VAR) (Equation (5)):(5)$$\%VAR=\left[1-\frac{S_{xy}^{2}}{S_{y}^{2}}\right]*100$$
where *S_xy_* is the standard deviation of the estimation; *S_y_* is the standard deviation of the sample.

The effect of temperature on the effective diffusivity was described by an Arrhenius type equation: (6)$$D_{eff}=D_{0}exp\left(-\frac{E_{a}}{RT}\right)$$
where *D*_0_ is a pre-exponential factor of the Arrhenius equation (m^2^/s); *E_a_* is the activation energy for water diffusion (kJ/mol); *R* is the constant of ideal gases (kJ/mol K); *T* is the temperature (K).

### 2.4. Dried Product Characterization

The contents of alcohol insoluble residue (AIR) and some antioxidant properties were measured for the purposes of studying the influence of the drying conditions on the obtained dried products.

#### 2.4.1. Alcohol Insoluble Residue (AIR)

The AIR of the dried samples was obtained by successive washing cycles of the dried samples in boiling ethanol (85% *v*/*v*) and pure acetone, as described by Femenia et al. [17]. After that, the swelling capacity (SC), water retention capacity (WRC) and fat adsorption capacity (FAC) of AIR were determined.

In the case of SC determination, 0.20 ± 0.01 g of AIR was placed into a measuring cylinder with 10 mL of distilled water. After 24 h of resting at room temperature (20 ± 1 °C), the volume occupied by the sample was determined and expressed as mL/g AIR.

As for the WRC estimation, 0.20 ± 0.01 g of AIR were placed into a 20 mL centrifuge tube with 10 mL of distilled water and kept for 24 h at room temperature (20 ± 1 °C). Then, it was centrifuged (Medifriger BL-S, P. Selecta, Barcelona, Spain) at 6000 rpm and 4 °C for 15 min. The supernatant was decanted and the precipitate weighed. The difference between the initial and final weight was the WRC and it was expressed as g water/g AIR.

Finally, for the FAC measurement, 0.20 ± 0.01 g of AIR were placed into a 20 mL centrifuge tube with 10 mL of sunflower oil. After 24 h at room temperature (20 ± 1 °C), the mix was subsequently centrifuged at 6000 rpm and 4 °C for 15 min. After removing the supernatant, the final weight was measured. The difference between the initial and final AIR weights was the absorbed oil and it is a measurement of the FAC. It was expressed as g oil/g AIR.

#### 2.4.2. Antioxidant Properties

Ethanolic extracts were obtained by mixing 1.00 ± 0.02 g of dried sample with 20 mL of ethanol 96% and homogenized for 1 min with an ultra-Turrax (T25 Digital, IKA, Staufen, Germany) at 13,500 rpm. The mixture was maintained at 4 °C for 24 h. Next, it was centrifuged at 4000 rpm and 4 °C for 10 min. Then, it was filtered using a 1.2 μm glass microfiber filter grade (GFFC Prat Dumas, Couze-et-Saint-Front, France) and made up to 20 mL with pure ethanol. Finally, the samples were stored in the dark at 4 °C until analysis [21].

Total Phenolic Content (TPC): The TPC was determined following the method described by Gao et al. [22] using the Folin-Ciocalteu reagent. Each trial was performed in triplicate. An aliquot of 0.1 mL of extract was homogenized with 0.2 mL of Folin-Ciocalteu reagent and 2 mL of distilled water. This mixture was kept at room temperature (20 ± 1 °C) for 3 min. Then, 1 mL of 20% Na_2_CO_3_ (*w*/*v*) was added, homogenized and maintained in the dark for one hour at room temperature. Finally, the absorbance of the samples at 765 nm was measured in a spectrophotometer (Helios Gamma, Thermo Spectronic, Cambridge, UK). The TPC was determined using a calibration curve built with a known concentration of gallic acid. The results were expressed as milligrams of gallic acid equivalent per gram of dry matter (mg GAE/g d.m.).Antioxidant Capacity (AC): The AC was measured by the FRAP (Ferric Reducing Antioxidant Power) method [23]. This method is based on the power of an antioxidant substance to reduce the 2,4,6-Tri(2-pyridyl)-s-triazine (TPTZ) ferric complex, which is colorless, to a ferrous complex, which is blue in color. This difference is measured from the determination of maximum absorbance at 595 nm. The FRAP method requires a previous preparation of 0.3 M anhydrous sodium acetate buffer pH 3.6; FeCl_3_ 6H_2_O 20 mM; and TPTZ 10 mM in HCL 40 mM. Subsequently, the FRAP reagent was prepared by mixing 10 mL of the buffer, 10 mL of the TPTZ solution and 10 mL of the FeCl_3_ solution and leaving it for 30 min in a bath (Tecton 200, P-Selecta, Barcelona, Spain) at 37 °C. Then, 30 µL of distilled water was added to a disposable cell. Next, 30 μL of bract extract (or ethanol in the case of the blank) was added. In the case of the stems, it was necessary to use a different proportion: 7.5 μL of sample and 22.5 μL of ethanol (1:4 dilution). Finally, 900 μL of the FRAP reagent was added and cells were placed into a 37 °C bath for 30 min. Finally, the absorbance was measured on a spectrophotometer (Helios Gamma, Thermo Spectronic, Cambridge, UK) at 595 nm. The results were expressed in μmol TROLOX/g d.m.Vitamin C (VC): The VC was estimated by determining the content of ascorbic acid using the method proposed by Dani and Jagota [24] with slight modifications. For this purpose, 0.5 mL of sample extract was mixed with 0.5 mL of a 7.5% solution of trichloroacetic acid. The solution was homogenized, maintained for 5 min at 4 °C and filtered. Then, 0.2 mL of the prepared extract, 2 mL of distilled water and 0.2 mL of a diluted solution (1:10 *v*/*v*) of the Folin-Ciocalteu reagent were placed into a spectrophotometric cell. After 10 min at room temperature, the absorbance at 760 nm was measured. A calibration curve was prepared with ethanolic solutions of known concentrations of ascorbic acid. The results were expressed as milligrams of ascorbic acid per gram of dry matter (mg VC/g d.m.).

### 2.5. Statistical Analysis

The results obtained were statistically analyzed using an analysis of variance (ANOVA, *p* < 0.05) and the significance of the differences between treatments was established with the LSD test (Least Significant Difference) intervals using the Statgraphics Centurion XVI (version 16.1.17) software (The Plains, VA, USA).

## 3. Results

### 3.1. Drying Kinetics and Modeling

#### 3.1.1. Influence of Drying Temperature

The initial moisture content of the bracts was 4.7 ± 0.4 kg water/kg dry matter (d.m.) and 6.3 ± 0.5 kg water/kg d.m. in the case of the stems.

As expected, the air dying temperature had a significant (*p* < 0.05) influence on the drying kinetics of both by-products (Figure 1): the higher the air temperature, the shorter the drying time required. The greater energy in the system provided by the hotter drying air increased the vapor pressure inside the sample, which caused greater water molecule movement and more water evaporation. This effect of drying temperature has been previously observed when drying other by-products. This is the case of apple peel, where reductions of 23% were achieved by increasing the temperature from 30 to 70 °C [25], or the by-products of cumbeba waste, which presented drying time reductions of 50% when the temperature was raised from 50 to 80 °C [26]. 

At the same temperature, drying time was different for both by-products, bracts drying significantly faster than the stems (Table 1). Thus, in the experiments carried out at the lowest temperature tested, 40 °C, the time needed to reach a moisture content of 0.25 ± 0.01 kg water/kg d.m. in bracts was 33% shorter than the time needed in stems. These differences were lightly reduced with the increase of drying temperature: drying time at 120 °C was 25% shorter in bracts compared with stems.

#### 3.1.2. Modeling

As previously explained, unidimensional moisture mass transport was assumed in samples for modeling purposes. For this, an infinite slab geometry was considered for the bracts and an infinite cylinder for the stems. For both kinds of samples, the proposed diffusion model fitted adequately, as shown by the percentage of variance explained, in over 99% of all cases (Table 1).

The effective diffusivity (*D_eff_*) identified ranged from 5 × 10^−10^ to 34 × 10^−10^ m^2^/s for the bracts and from 3.6 × 10^−10^ to 6.1 × 10^−10^ m^2^/s for the stems. These values are in the range of those reported for coffee [27], quince [28] or blueberries [16]. The fact that the *D_eff_* values of the bracts are greater than those of the stems indicates a faster internal moisture movement, which can be attributed to the different tissue structure.

The drying temperature significantly affected the *D_eff_* (*p* < 0.05). Thus, in the case of the bracts, the identified value in experiments carried out at 120 °C was 580% higher than that obtained at 40 °C. For the stems, this increase was 69%. For both by-products, the temperature influence on *D_eff_* presented an Arrhenius type relationship (Equation (6)), in both cases achieving a correlation coefficient of over 99%. The activation energy obtained was 24.9 and 7 kJ/mol for bracts and stems, respectively. The lower *E_a_* value of the stems indicated a milder influence of the drying temperature on the drying rate, since this parameter depends on the components, tissue structures and specific surface area of the samples [29]. Therefore, from a kinetics point of view, the rise in the drying temperature is more effective at accelerating the drying of the bracts than that of the stems. In any case, due to the fact that a rise in the drying temperature leads to an increase in the energy consumption but a shortening of the drying time, it is necessary to study the influence on the total energy requirements of the process.

As to the identified values of the mass transfer coefficient (k), they were also greater in the bracts than in the stems. In this case, the different sample geometry, either slabs or cylinders, can affect how the air circulates around them and may explain this difference. The temperature also affected the values of k: the higher the air temperature, the higher the values in both cases. Thus, an increase in the air temperature implies faster heat transport between the drying air and the samples, which permits quicker moisture vaporization and transport from the sample surface to the surroundings.

In summary, the internal (*D_eff_*) and external (k) moisture transport in bracts is faster than in stems; moreover, it is more sensitive to the increase of drying temperature. Therefore, based on these results and from a kinetics point of view, the drying optimization of bracts and stems requires that both products are separately dried.

### 3.2. Influence of Drying Conditions on Product Characteristics

Not only was the influence of the drying temperature quantified in terms of the drying kinetics, but also in terms of the final product properties, such as the alcohol insoluble residue and its characteristics, or the antioxidant potential of both dried by-products.

#### 3.2.1. Alcohol Insoluble Residue (AIR)

The AIR content represents a measurement of the content of insoluble fiber. The results obtained in samples of fresh bracts and stems were 0.85 kg/kg d.m. and 0.63 kg/kg d.m., respectively. These values were in the range of those reported by Domingo et al. [30] and Femenia et al. [17], also working on artichoke bracts and stems. Drying significantly reduced the AIR content; this reduction becoming greater the higher the drying temperature rose. Thus, for both the bracts and stems, the maximum AIR values were obtained at 40 °C (Figure 2). Above this temperature, these values were observed to decrease in line with the temperature in the case of the bracts. As to the stems, this reduction occurred up to a drying temperature of 80 °C, and higher temperatures did not significantly affect the AIR values obtained.

The drying process could affect the structural disposition of the cell wall polysaccharides. In order to evaluate these possible modifications, the swelling capacity (SW), water retention capacity (WRC) and fat absorption capacity (FAC) were determined (Figure 3). As for the SW, the values obtained for the AIR of stems were significantly greater than those of the bracts. The drying temperature affected the SW of both by-products differently. Thus, while for the bracts, an increase in the drying temperature produced a significant decrease (*p* < 0.05) in the SW values, in the case of the stems, the SW values were higher in those samples dried at intermediate temperatures, exhibiting a maximum value at 100 °C (Figure 3a). This difference could be attributed to the different composition of both by-products [17], since the SW is related to the content on polyscaccharides [8], and their response to the temperature. Thus, bracts composition could include more sensitive components to temperature than in the case of stems.

As for WRC, the values observed for the stems were also higher than for the bracts at every temperature tested. The increase in the air drying temperature led to a decrease in the WRC values (Figure 3b) in both cases. The increase in air temperature may lead to cellular damage of the main polysaccharides affecting the ability of retain water. This trend has also been found by Garau et al. [31] for AIR from orange peel and pulp.

Finally, FAC figures obtained were in the range that the reported by Boubaker et al. [8] and were also higher in the AIR of stems than in that of the bracts at every drying temperature tested. In both cases, it was observed that the highest FAC was obtained at the moderate drying temperature (60 °C); at higher and lower temperatures, it was lower (Figure 3c).

Therefore, although the stems had a lower AIR content than the bracts, the SW, WRC and FAC values were significantly higher. As far as the drying temperature is concerned, the most appropriate values for these variables were found at intermediate temperatures (60–80 °C) in both by-products. Similar drying temperature influence was reported by Garau et al. [31] in the case of orange peel.

#### 3.2.2. Antioxidant Properties

The influence of drying on the antioxidant properties of the artichoke stems and bracts was studied from the determination of the total phenolic content (TPC), the antioxidant capacity (AC) and the vitamin C content (VC).

The TPC of the fresh bracts was 0.034 ± 0.003 mg GAE/g d.m. Regarding the value measured in stems, it was one order of magnitude greater than the obtained in bracts, specifically 0.48 ± 0.04 mg GAE/g d.m. The higher TPC content in stems than in bracts has been previously reported by Lombardo et al. [18] or Zuorro et al. [32].

Drying reduced these values significantly (*p* < 0.05) (Table 2). This decrease can be attributed to the union of phenolic compounds with other compounds (proteins) or to alterations in their chemical structure. In this sense, when studying the drying of pears, Djendoubi Mrad et al. [33] observed that the TPC decreased linearly with the drying time. Thus, a 30% reduction in TPC in comparison with fresh pear was observed in pear dried for 10 h at 30 °C. Fratianni et al. [34] also reported a decrease in phenolic compounds during mango drying at 50 and 70 °C.

As to the influence of drying temperature, the highest TPC value of the dried bract samples was found when the drying took place at 40 °C. Higher drying temperatures (60, 80, 100 and 120 °C) produced a lower TPC with no significant differences (*p* > 0.05) between them. In the case of the stems, the effect of temperature was the opposite; a slight increase in the TPC content was obtained at the highest temperatures tested, i.e., 100 and 120 °C. Rodríguez et al. [35] found a TPC reduction ranging from 20–39% of the TPC of fresh apple after drying at 30, 50 and 70 °C. Studying the convective drying of carrot peel, Chantaro et al. [36] also observed a 64% loss in TPC when drying at 60 °C and 26% when drying at 80 °C. Garau et al. [31] reported that the longer drying times needed at low temperatures resulted in a reduction in the TPC of orange by-products (drying temperatures ranged from 30–90 °C). Therefore, as drying is a thermal treatment, the degradation of the phenolic compounds can be attributed to the combined effect of drying time and drying temperature.

As to the AC of the fresh bracts and stems, the values obtained were 0.34 ± 0.03 and 0.37 ± 0.03 mg TROLOX/g d.m. respectively. Noriega-Rodríguez et al. [19] attributed the AC of artichoke mainly to the phenolic compounds present in the product. In this study, a strong correlation of AC with the TPC has also been found (0.8801 and 0.7509, for bracts and stems, respectively). In this way, drying also significantly reduced (*p* < 0.05) the AC values if compared to fresh samples (Table 2). Fratianni et al. [34] also reported a decrease in the antioxidant activity when drying mango at 70 °C. Similarly to TPC, the AC was significantly higher (*p* < 0.05) in the bract samples dried at 40 °C than in those dried at the higher temperatures, although the differences were small. In the case of the stem samples, the rise in the drying temperature also increased the AC of the samples. Thus, the AC value of the samples dried at 120 °C was 358% greater (*p* < 0.05) than those dried at 40 °C. The fact that there is a greater amount of AC in the stems when drying at high temperatures could be due to the generation of new compounds with a high antioxidant activity via the reaction between existing compounds [37]. During the drying of apples, Rodríguez et al. [35] found depletions in the antioxidant activity of 51.2 ± 2.2%, 45.6 ± 2.0% and 38.1 ± 2.3% at drying temperatures of 30, 50 and 70 °C, respectively.

Finally, drying also reduced the vitamin C content under every condition tested (Table 2), which could be attributed to the oxidation reactions. The reduction in VC during drying has been found in peppers [38] or cantaloupe juice powders [39]. In the case of the bracts, the highest VC was obtained in the samples dried at 40 °C. As for the stems, greater retention was observed at 120 °C; a likely reason for this could be the long drying time required at low and intermediate temperatures to reach the final moisture content, which can favor vitamin C degradation reactions. Similar results have been found by Jin et al. [40] and Roshanak et al. [41] in the drying of broccoli and tea, respectively. They found that both factors, temperature and drying time, affected the oxidation of vitamin C and, therefore, its content in the final products.

The aim of by-product drying is simply to achieve its stabilization while preserving its properties. Therefore, according to the obtained results, the best drying temperature for bracts was 40 °C since it provided the highest values of both SW, WRC, FAC and also of the antioxidant properties. On the contrary, in the case of the stems, the best temperature of those tested was 120 °C, where the highest values of fiber and antioxidant properties were obtained.

## 4. Conclusions

The increase of drying temperature shortened the drying process of both artichoke by-products, bracts and stems. The stems drying was slower and clearly exhibited lower activation energy than the bracts, which meant a milder temperature influence on the drying rate. Both by-products presented relatively large amounts of AIR, this being greater in the bracts. However, the AIR of the stems presented higher values of SW, WRC and FAC at every temperature tested. For the drying conditions tested, the TPC, AC and VC values of the dried samples decreased significantly compared to the fresh samples (*p* < 0.05). The influence of the drying temperature properties was quite different for the two by-products, with the best values found at 40 and 120 °C for the bracts and the stems, respectively. This suggests that the two main by-products of the artichoke should be dried separately to improve drying and obtain the best possible quality in each, which makes possible their use for example as a fiber or antioxidant supplement.

## Figures and Tables

**Figure 1 foods-10-00459-f001:**
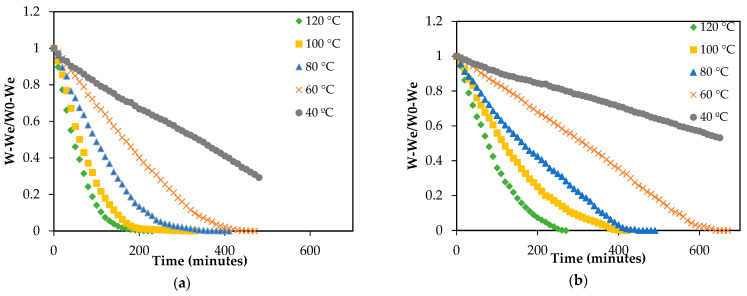
Evolution of the dimensionless moisture content during drying of artichoke bracts (**a**) and stems (**b**) at different temperatures.

**Figure 2 foods-10-00459-f002:**
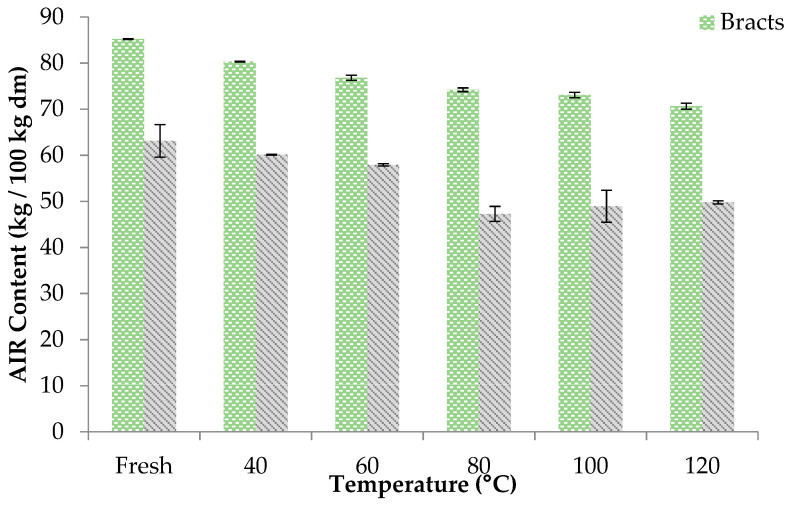
Alcohol insoluble residue (AIR) content of bracts and stems samples after to be dried at different temperatures.

**Figure 3 foods-10-00459-f003:**
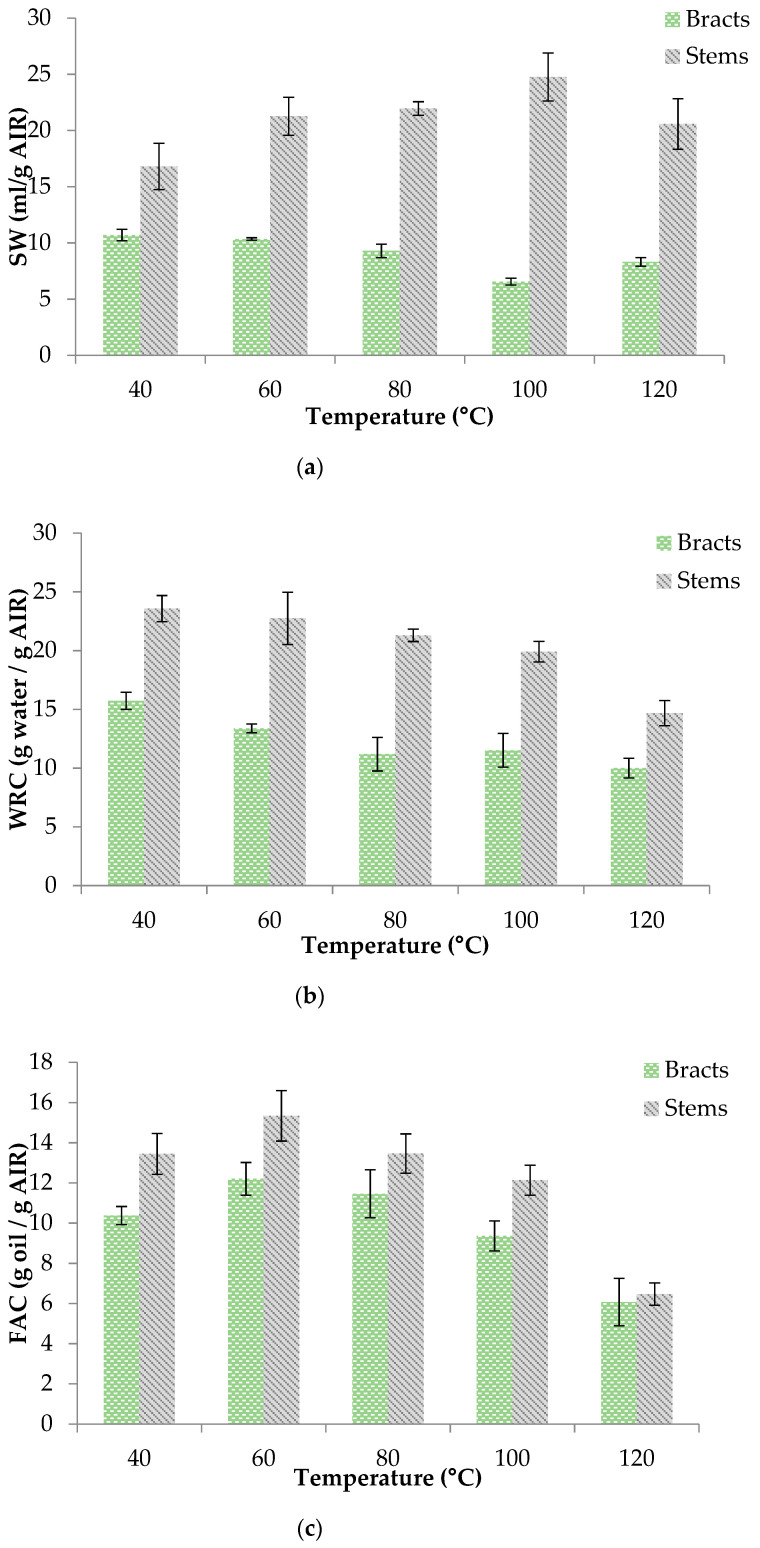
AIR functional properties of bracts and stems: (**a**) Swelling Capacity (SW); (**b**) Water Retention Capacity (WRC); (**c**) Fat Absorption Capacity (FAC).

**Table 1 foods-10-00459-t001:** Drying experiments of bracts and stems of artichoke at different temperatures. Drying time (h) needed to reach a moisture content of 0.25 kg W/kg dm. Effective diffusivity (*D_eff_*) and mass transfer coefficient (k) (average values and standard deviation) identified from modeling and percentage of variance (% VAR) explained by the model.

	Drying Temperature (°C)	Drying Time (h)	*D_eff_* (10^−10^ m^2^/s)	k (10^−4^ m^2^/s)	% VAR
Bracts	40	12.0 ± 0.3 ^a^	5.0 ± 0.4 ^a^	4.1 ± 0.2 ^a^	99.5 ± 0.8
	60	6.00 ± 0.17 ^b^	8.8 ± 0.3 ^b^	4.3 ± 0.1 ^b^	99.5 ± 0.2
	80	5.00 ± 0.15 ^c^	15 ± 5 ^c^	13.6 ± 0.6 ^c^	99.8 ± 0.3
	100	4.0 ± 0.2 ^d^	25 ± 4 ^d^	26 ± 5 ^d^	99.8 ± 0.4
	120	3.0 ± 0.2 ^e^	34 ± 4 ^e^	47 ± 4 ^e^	99.6 ± 0.2
Stems	40	18.0 ± 0.6 ^j^	3.6 ± 0.3 ^f^	1.3 ± 0.8 ^f^	99.6 ± 0.2
	60	10.0 ± 0.4 ^g^	4.1 ± 0.5 ^fg^	2.2 ± 0.3 ^fg^	99.1 ± 0.1
	80	7.0 ± 0.3 ^h^	4.6 ± 0.2 ^h^	4.3 ± 0.2 ^gh^	99.2 ± 0.7
	100	6.00 ± 0.14 ^i^	5.4 ± 0.3 ^hi^	5 ± 1 ^h^	99.0 ± 0.1
	120	4.0 ± 0.2 ^j^	6.1 ± 0.6 ^i^	7.1 ± 0.2 ^i^	98.7 ± 0.3

Different letters in the same column indicate significant differences according to an LSD test (*p* < 0.05).

**Table 2 foods-10-00459-t002:** Total Phenolic Content (TPC), vitamin C content (VC) and Antioxidant Capacity (AC) of artichoke bracts and stems, fresh and dried at different temperatures.

	Drying Temperature (°C)	TPC (mg GAE/g dm)	VC (mg VC/g dm)	AC (mg TROLOX/g dm)
Bracts	-	0.034 ± 0.003 ^a^	0.46 ± 0.05 ^a^	0.34 ± 0.03 ^a^
	40	0.022 ± 0.003 ^b^	0.13 ± 0.02 ^b^	0.09 ± 0.01 ^b^
	60	0.009 ± 0.003 ^c^	0.06 ± 0.02 ^c^	0.05 ± 0.01 ^c^
	80	0.009 ± 0.002 ^c^	0.03 ± 0.01 ^d^	0.039 ± 0.005 ^d^
	100	0.008 ± 0.002 ^c^	0.04 ± 0.01 ^d^	0.031 ± 0.003 ^d^
	120	0.008 ± 0.002 ^c^	0.07 ± 0.02 ^c^	0.032 ± 0.006 ^d^
Stems	-	0.48 ± 0.04 ^d^	2.0 ± 0.1 ^d^	0.37 ± 0.03 ^e^
	40	0.04 ± 0.01 ^e^	0.12 ± 0.05 ^e^	0.08 ± 0.02 ^f^
	60	0.04 ± 0.01 ^e^	0.04 ± 0.02 ^f^	0.07 ± 0.02 ^g^
	80	0.03 ± 0.01 ^e^	0.08 ± 0.03 ^df^	0.06 ± 0.02 ^g^
	100	0.07 ± 0.01 ^f^	0.36 ± 0.06 ^g^	0.21 ± 0.02 ^h^
	120	0.09 ± 0.01 ^g^	0.55 ± 0.06 ^h^	0.23 ± 0.02 ^h^

Different letters in the same column indicate significant differences according to an LSD test (*p* < 0.05).

## Data Availability

Data is contained within the article.

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
