# Peer review of "Artichoke by Products as a Source of Antioxidant and Fiber: How It Can Be Affected by Drying Temperature"

_foods, 2021, doi:10.3390/foods10020459_

Round 1
Reviewer 1 Report
Development of processes to achieve the higher utilization ratio of food industry by-products present great opportunity for food industry to achieve higher resource efficiency and conform with the main principles of circular economy, as well. Manuscript foods-1117652 focuses on the investigation of drying parameters on phenolic content antioxidant capacity and vitamin C content of artichoke by-products. Manuscript is generally well structured. Method of drying kinetic modelling, antioxidant properties and AIR measurements are given in details. Manuscript contains valuable results discussed with relevant references. But, in my opinion, manuscript needs revision.
Comments, suggestions:
I suggest the authors to discuss briefly the applicable drying methods and drying kinetic models in Introduction section.
It should be highlighted how was chosen the drying parameteres 8based on literature, preliminary experiments or modelling the industrially used parameters, for instance).
Figure 1 a) and b) do not contain the errors/deviations.
Authors concluded in line 270-274 that ’ The drying temperature affected the SW of both by-products differently. Thus, while for the bracts, an increase in the drying temperature produced a significant decrease (p<0.05) in the SW values, in the case of the stems, the SW values werehigher in those samples dried at intermediate temperatures, exhibiting a maximum value at 100 ºC (Figure 3a)’. But the reason of the decreasing effect of temperature higher than 100 Celsius is not discussed. Please add explanation for this phenomena (heat sensitive components. structural change etc, for instance).
Author Response
foods-1117652
Artichoke by products as a source of antioxidant and fiber. How it can be affected by drying temperature
Response to Reviewer #1:
Thank you very much for your valuable comments. We have carefully considered all of them and revised the manuscript accordingly. Below you can find the responses point-by-point. The recent revised contents have been marked in red in the manuscript.
The following considerations have been made:
Comment reviewer: I suggest the authors to discuss briefly the applicable drying methods and drying kinetic models in Introduction section.
Answer: As suggested by reviewer, a comment about the drying method and the drying kinetics models has been added in the introduction section.
New version, lines 46-60: “The most commonly used drying technique is hot air drying [13] due to its simplicity and the ease control of process compared with other more complex or expensive techniques. In this case, the water transport from a material fundamentally depends on the external conditions of temperature, relative humidity, velocity and the direction of the hot air flow, as well as the characteristics and geometry of the solid [14]. Modelling of drying kinetics permits the objective quantification of the influence of process variables in the rate of the process and it is essential to optimize the operation. The use of theoretical models, such as those based in Fick’s law of diffusion, is preferred than the empirical ones because they can be useful to predict moisture content evolution in different conditions they were fitted, e. g. different sample geometry or initial moisture content.
The most used technique to intensify hot air drying is the increase of drying temper-ature. However, the influence of temperature in drying rate depends on the products and sometimes the shortening of drying time could not compensate the increase of energy consumption, which the increase of temperature means. Moreover, …”
Comment reviewer: It should be highlighted how was chosen the drying parameteres 8based on literature, preliminary experiments or modelling the industrially used parameters, for instance).
Answer: Following the reviewer suggestion, this fact has been clarified in the new version of manuscript
New version, lines 84-89: “The air drying tests were performed at atmospheric pressure using an air velocity of 1 m/s and at five different temperatures (40, 60, 80, 100 and 120 °C). This range of temperature and air velocity is usually applied in the drying of agro-food products [13, 14, 15]. The relative humidity remained below 10% in every experiment carried out. At pre-set intervals (10 min), samples were taken out from the chamber and weighed.”
Comment reviewer: Figure 1 a) and b) do not contain the errors/deviations.
Answer: The figure shows the average moisture content evolution observed for bracts and stems at the different drying temperatures tested. In both cases, the replicates of each drying conditions showed a quite repetitive behaviour. Therefore, the error bars were very small and only were superimposed just at the beginning of the experiments. Thus, drying kinetics determined at the temperatures tested were significantly different. This fact has been clarified in the text (Line 201-202: “As expected, the air dying temperature had a significant (p<0.05) influence on the drying kinetics of both by-products (Figure 1)”). The inclusion of the error bars in the figure 1a and 1b does not contribute to better show this fact. On the contrary, it makes the figure less clear due to the high number of data included and the small size of the error bars. For this reason, we prefer do not include them.
Comment reviewer: Authors concluded in line 270-274 that ’ The drying temperature affected the SW of both by-products differently. Thus, while for the bracts, an increase in the drying temperature produced a significant decrease (p<0.05) in the SW values, in the case of the stems, the SW values werehigher in those samples dried at intermediate temperatures, exhibiting a maximum value at 100 ºC (Figure 3a)’. But the reason of the decreasing effect of temperature higher than 100 Celsius is not discussed. Please add explanation for this phenomena (heat sensitive components. structural change etc, for instance).
Answer: According to the reviewer, a short explanation it has been added.
New version, line 291-294: “This difference could be attributed to the different composition of both by-products [17], since the SW is related to the content on polyscaccharides [8], and their response to the temperature. Thus, bracts composition could present components more sensitive to higher temperatures than in the case of bracts.”

Reviewer 2 Report
Since the work is focusing on the processing of by-products which are usually not used in diet but discarded, would be interesting to know which application could they found which would justify and add value to the research work which was carried out. In this regard, both introduction and conclusion can be improved.
Author Response
foods-1117652
Artichoke by products as a source of antioxidant and fiber. How it can be affected by drying temperature
Response to Reviewer #2: Thank you very much for your valuable comments. We have carefully considered all of them and revised the manuscript accordingly. Below you can find the responses point-by-point. The recent revised contents have been marked in red in the manuscript.
The following considerations have been made:
Comment reviewer: Since the work is focusing on the processing of by-products which are usually not used in diet but discarded, would be interesting to know which application could they found which would justify and add value to the research work which was carried out. In this regard, both introduction and conclusion can be improved.
Answer: The main applications of artichoke by-products can be found in Introduction section (line 38-41). Moreover, as the reviewer suggest, some sentences have been added/modified in the Introduction and the Conclusion section to clarify this point.
New version, line 66-70: “Therefore, the objective of this study was to separately assess the influence of the drying temperature on the drying kinetics and on some quality related parameters (fiber and antioxidant properties) of both artichoke by-products, external bracts and stems. In this sense, it could be identify the better drying conditions, which can improve their uses as functional ingredient or fiber/antioxidant supplement.”
New version, Line 396-398: “This suggests that the two main by-products of the artichoke should be dried separately to improve drying and obtain the best possible quality in each, which make possible their use for example as a fiber or antioxidant supplement”

Reviewer 3 Report
Dear Authors,
The article presented is original and makes contributions of interest regarding Food Engineering and Technology. Research methodology is very well described. The aim of the article has been accomplished and the conclusions are well supported by results. The number of bibliographic items is numerically sufficient and sufficiently varied.
However I ‘m recommending some revisions to your manuscript.
Materials and methods
- 1 Raw material: More info should be provided regarding the sampling (for example the number of samples, ecc.)
Conclusions
- Line 366-367: This sentence is not clearly presented. Rewrite
Author Response
foods-1117652
Artichoke by products as a source of antioxidant and fiber. How it can be affected by drying temperature
Response to Reviewer #3: Thank you very much for your valuable comments. We have carefully considered all of them and revised the manuscript accordingly. Below you can find the responses point-by-point. The recent revised contents have been marked in red in the manuscript.
The following considerations have been made:
Comment reviewer: Materials and methods
- 1 Raw material: More info should be provided regarding the sampling (for example the number of samples, ecc.)
Answer: More information about the sampling has been added to the manuscript
New version, line 77-82: “The bracts and stems were separated from the artichokes to simulate the waste produced by the industrial processing of artichokes and independently processed. In the case of bracts, only the external ones were considered extracting a number of 20-25 bracts from each artichoke. The stems were transversally cut to obtain homogeneous cylindrical samples of 15.0 ± 0.5 mm in height (3-4 by artichoke).
Comment reviewer: Conclusions
- Line 366-367: This sentence is not clearly presented. Rewrite
Answer: The sentence has been rewritten
New version, line 385-386: “The increase of drying temperature shortened the drying process of both artichoke by-products, bracts and stems.”

Reviewer 4 Report
The authors of the review entitled "Artichoke by products as a source of antioxidant and fiber. How it can be affected by drying temperature " described their work related to the drying effects on the waste products of artichoke and Mediterranean food. The manuscript is well written, and the results are successfully presented in figures. The research group is strong, and the investigation was well developed. The introduction from my point of view is short and must be extended, explaining a little more about the waste process and how this waste can be utilized. The work is in great shape and I only missed more detail about the experimental conditions. What about the pressure, humidity will affect? Should be interesting if the authors analyze the subproducts in each temperature. They stated in the conclusion that the best temperatures are 40 and 120 degrees. Are those byproducts similar? Or totally different. A short discussion should good. Any idea of the subproducts? I also found a couple minor stuff.
- Page 1, line 30 italics. Please use italics in the scientific names even in the reference section.
- Page 2, line 77. Missing unit
- Be consistent on the degree symbol through the manuscript sometimes they underline the symbol. Is it means something? For example, page 4, line 140, lines 192, 195, etc…
Author Response
foods-1117652
Artichoke by products as a source of antioxidant and fiber. How it can be affected by drying temperature
Response to Reviewer #4: Thank you very much for your valuable comments. We have carefully considered all of them and revised the manuscript accordingly. Below you can find the responses point-by-point. The recent revised contents have been marked in red in the manuscript.
Comment reviewer:
The introduction from my point of view is short and must be extended, explaining a little more about the waste process and how this waste can be utilized. The work is in great shape and I only missed more detail about the experimental conditions. What about the pressure, humidity will affect? Should be interesting if the authors analyze the subproducts in each temperature. They stated in the conclusion that the best temperatures are 40 and 120 degrees. Are those byproducts similar? Or totally different. A short discussion should good. Any idea of the subproducts? I also found a couple minor stuff.
Answer: The introduction section has been extended in the new version including information about drying process, drying modelling and artichoke by products uses and possible application. See new version
Comment reviewer:
What about the pressure, humidity will affect?
Answer: The aim of the paper was to study the influence of drying temperature in the drying process and in the quality of the obtained product. For this reason, the other process variable was maintained constant for every experiment. Thus, all experiments were carried out at atmospheric pressure. Regarding the relative humidity of drying air, it was always bellow 10 %. Com changes have been introduced in the text to clarify this.
New version, line 84-87: “The air drying tests were performed at atmospheric pressure using an air velocity of 1 m/s and at five different temperatures (40, 60, 80, 100 and 120 °C). This range of temperature and air velocity is usually applied in the drying of agro-food products [13, 14, 15]. The relative humidity remained below 10% in every experiment carried out.”
Comment reviewer:
They stated in the conclusion that the best temperatures are 40 and 120 degrees. Are those byproducts similar? Or totally different. A short discussion should good. Any idea of the subproducts?
Answer: Bracts and stems have a different structure but usually are jointly processed. However, the results showed that the behavior of both by-products regarding drying temperature was quite different and for this reason, it is recommend processing them separately. Some suggestions about the use of these by products have been included in the manuscript
Comment reviewer:
Page 1, line 30 italics. Please use italics in the scientific names even in the reference section.
Page 2, line 77. Missing unit
Be consistent on the degree symbol through the manuscript sometimes they underline the symbol. Is it means something? For example, page 4, line 140, lines 192, 195, etc…
Answer: All this mistakes have been corrected in the new version of the manuscript
